# Runoff–Sediment Simulation of Typical Small Watershed in Loess Plateau of China

**Shengqi Jian [1,2], Peiqing Xiao [2,*], Yan Tang [1] and Peng Jiao [2]**

1   Yellow River Laboratory, Zhengzhou University, Zhengzhou 450001, China
2   Key Laboratory of Soil and Water Conservation on the Loess Plateau of Ministry of Water Resources, Yellow River Institute of Hydraulic Research, Zhengzhou 450003, China
*   Correspondence: peiqingxiao@163.com

**Abstract:** The implementation of measures such as check dams and terraces in the Loess Plateau of China has had a groundbreaking impact on water and sediment conditions. The question of how to accurately simulate the runoff–sediment process under complex underlying surface conditions has become key to clarifying the water cycle law. This study focused on the Chenggou River basin, a small watershed located in the Loess Plateau, to examine the effect of the underlying surface characteristics on the runoff production process, and the spatial distribution of the dominant runoff process in the runoff generation mechanism was determined according to the land application, slope and vegetation coverage of the watershed. A runoff–sediment model was constructed that was combined with the traditional hydrological physical mechanism and a deep learning algorithm. Different parameters were calibrated depending on the spatial distribution of the dominant runoff process and we then ran the runoff–sediment simulation model to very its serviceability in the typical watershed of the Loess Plateau. Different parameters were calibrated for each type of hydrological response unit (HRU), according to the division of each HRU and the actual flood process, to calculate the runoff yield of each HRU. An LSTM algorithm was used for flow routing and a CSLE algorithm was used to simulate soil erosion. The results show that there were 29 flood events in the Chenggou River basin from 2013 to 2017. The average runoff depth had an 8.86% margin of error, while the peak flow had a slightly higher 9.44% deviation. The Nash efficiency coefficient was 0.84, and the relative error of soil erosion was 14.45%. The model simulation effect is good and can be applied to the typical watershed of the Loess Plateau. The model can provide a scientific basis for the highly efficient and sustainable utilization of water resources, ecological environment construction and the sustainable development of agriculture.

**Keywords:** check dams; terraces; runoff generation mechanism; runoff–sediment simulation; deep leaning

## 1. Introduction

The problem of runoff sediment in the Yellow River, Gansu province, China has always been a concern. The problem of sediment deposition caused by high sediment concentration in the river has remained unsolved for a long time, resulting in frequent flood disasters. There are different degrees of flood hazards along the coast [1,2]. The Loess Plateau, a major contributor to the sediment in the Yellow River and a hotspot for soil erosion, has been the focus of Chinese environmental initiatives since the 1950s. These include national soil and water conservation projects, the Three-North Shelter Forest Program, natural forest protection, the conversion of farmland to forests and grasslands, check dams, and terrace construction [3]. By 2015, the number of check dams with sediment-retaining capacity on the Loess Plateau was 56,422. As of 2018, 36,897 m$^2$ of terraces have been built on the Loess Plateau [4] and vegetation coverage has also increased significantly [5]. Nestled in the Yellow River basin, the Loess Plateau has been the focus of a series of ecological initiatives. Owing to climate change, the flow and sediment discharge of the Yellow River

have seen a substantial decline since the 70s, particularly through the 90s, as demonstrated by recent studies [6].

The purpose of various ecological construction projects in the Loess Plateau has been to comprehensively control soil and water erosion and to improve the ecological outlook in the middle reaches. However, these measures are also considered to be the main reason for the significant changes in water and sediment. The change of underlying surface conditions directly affects the mechanism of runoff and sediment. Researchers throughout the globe have devoted much attention to the way that human activities and global warming are impacting runoff and sediment processes as well as their consequences [7]. To guarantee the lasting stability of the Yellow River basin, it is essential to sensibly regulate the water–sediment relationship, with the pivotal factor being a quantitative investigation of the effects of terrestrial alterations activated by human actions on runoff–sediment processes. The underlying surface conditions of the Loess Plateau have changed greatly, which has in turn changed the mechanism of runoff–sediment process in other related watersheds. Soil and water conservation measures can bring about a transformation to the surface of a watershed and, in turn, affect the runoff–sediment process. However, the extent of the impact is yet to be determined. Consequently, it is scientifically essential to examine the influence of such measures on the runoff–sediment process in watersheds in the Loess Plateau in order to deeply understand the law of runoff–sediment in the Loess Plateau, to strengthen the controllability of flood and sediment, and to improve the disaster response system and ability [8,9].

As the runoff and sediment of the Yellow River and its main tributaries have decreased to different degrees over the years, investigations into the precipitous dip in runoff and silt in the Yellow River basin have principally centered on the diverse soil and water conservation efforts carried out on the Loess Plateau, these include check dams and terraces, which constitute a considerable portion of the efforts [10–12]. The question of how to scientifically quantify the influence of check dams and terraces on water and sediment is of great practical significance to the direction of future projects for comprehensive soil and water conservation improvement, the regulation of water and sediment in the Yellow River basin, the improvement of ecological environment, and the high-quality development of regional economy [13,14]. Soil and water conservation measures can be evaluated using the widely recognized hydrological and water conservation methods. The hydrological method, in particular, is a quantitative approach to the assessment of the impact of precipitation and human involvement on the amount of runoff and sediment variation in watersheds. This is accomplished by establishing a link between rainfall, runoff and sediment during designated time frames [15]. The water conservation method involves superimposing the contribution of various factors and calculating their influence on the change of water and sediment. The development of runoff and sediment yield theory and calculation methods has increased recognition of the mechanisms and processes related to water and sediment. The hydrological model has become the go-to instrument for domestic and foreign scholars when calculating the efficacy of soil and water conservation actions in mitigating water and sediment losses [16]. Before the 1980s, limited by the theoretical basis and calculation level, scholars mainly put forward various conceptual hydrological models. These included the TANK model proposed by the National Disaster Prevention Center of Japan, the SAC model established by the Sacramento Forecast Center of the National Weather Service, and the California Department of Water Resources [17], and the Xin'anjiang model proposed by Chinese scholars on the basis of summarizing the empirical runoff generation and confluence calculation in many regions of China and analyzing the hydrological data of many years [18]. Li et al. (2015) [19] combined the Xin'anjiang model and the Hebei rain flood model, the flooding process in the Dongwan basin was simulated according to the underlying surface conditions which were divided into two types: infiltration and saturation excess runoff. Most hydrological models have only one runoff calculation method, and the unified parameters are generalized according to the whole watershed or sub-basin, as a result the hydrological calculation process is less precise. In the Loess

Plateau, the numerous soil and water conservation constructions that have been built are evident in their effects. The terrain is split into numerous pieces, making it hard to group the entire area into a single runoff generation pattern. It is necessary to use the hydrological response unit (HRU) as the calculation unit and consider the runoff calculation method of each HRU to improve the precision and the accuracy of the results [20,21].

Using different methods to explore the effects of check dams and terraces on water and sediment changes may lead to large differences. Gao et al. (2016b) [22] used the hydrological method to assess the effects of climate change and human activities on runoff and sediment in the Yanhe River basin from 1952 to 2011, which revealed a 67.13% and 80.10% reduction respectively. Xu et al. (2013) [23] compared the runoff–sediment production between 1956–1960, 1984–1987 and 2006–2008, where the check dam system intercepted 14.3% of the runoff and 85.5% of the sediment. Shi et al. (2016) [24] used the Yellow River digital watershed model to simulate the Huangfuchuan River basin's runoff from 1999 to 2012, which showed a 39% decrease due to the main dam. Li et al. (2017) [25] further used the SWAT model to simulate the same watershed's runoff from 2000 to 2012 and saw a 65.2% reduction. Sun et al. (2017) [26] conducted principal component analysis to analyze various factors' contributions to runoff and sediment changes in the Tuwei River from 1956 to 2010 and concluded that forest and grass had a greater effect than terraces and check dams. Contrarily, Zhao et al. (2019) [27] arrived at a conflicting result by employing the VAR model.

Studies on the alteration of the runoff–sediment mechanism in the Loess Plateau caused by environmental modification have been conducted, yet further research is needed to explore the changes of different underlying surface types. In particular, the hilly and gully region of the Loess Plateau has seen limited research on the impact of numerous check dams and terraces on the runoff–sediment watershed [28,29]. Many hydrological models have taken into account the different parameters under different underlying surface conditions but have rarely considered the different water and sediment mechanisms under different underlying surface conditions [30,31]. Although the simulation results show only a small number of erroneous deviations from the actual runoff yield of the watershed, the parameters may not necessarily meet its actual underlying surface conditions, resulting in different results being obtained by quantitative analysis of soil and water conservation measures using different methods. Thus, a hydrological model should be established in order to accurately and fairly evaluate the impact of soil and water conservation measures on water and sediment loss, taking into account the modification in runoff pattern.

The main contributions in the present study derive from watershed runoff theory. The discriminating mechanism of the predominant runoff process was established to identify the spatial dispersion of the dominant runoff across the watershed's underlying surface in order to ascertain the rationale behind the evolution of water and sediment processes in the study area at the mechanism level. Moreover, by taking into consideration the altering environment's spatial characteristics in terms of the runoff and sediment mechanism, the hydrological response unit of the watershed was divided, and a runoff–sediment model based on the conditional distribution of the underlying pad surface was constructed by amalgamating a traditional physical mechanism and deep learning.

## 2. Model Construction

### 2.1. Model Structure

To accurately identify the effect of soil and water conservation measures on runoff–sediment in the Loess Plateau, it is essential to differentiate the hydrological response unit from the alteration of the underlying surface, taking the features of various HRU into account, and examining the shift of runoff–sediment from a mechanistic point of view. It is generally believed that, for the Loess Plateau, infiltration excess runoff mainly occurs, which is determined by rainfall intensity and ground infiltration capacity. When the rainfall intensity exceeds the soil infiltration capacity, the watershed will produce runoff, and the yield will be the difference between the two. In cases where the soil infiltration capacity is greater than the

rainfall intensity, all water will infiltrate. However, for HRUs which are judged to be prone to saturation excess runoff, assessing the point of saturation and calculating the runoff for each period are key. In the model, we consider infiltration excess runoff during the initial rainfall phase when the soil water storage is full, followed by saturation excess runoff. In the confluence module, we used an artificial neural network—the long short-term memory neural network model (LSTM). The CSLE model based on individual rainfall was applied to the sediment yield module (Figure 1).

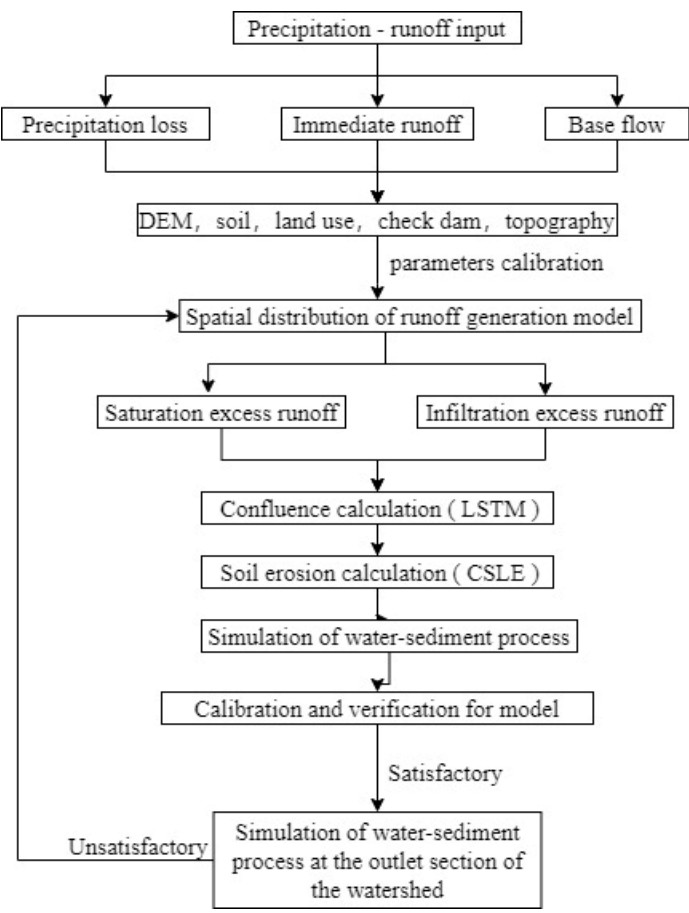

**Figure 1.** Model structure diagram.

### 2.2. Evapotranspiration Module

According to the location of evapotranspiration, we divided soil evaporation into upper and lower layers, and soil water consumption was carried out from top to bottom.

$$
\begin{aligned}
E_m &= kc \cdot E_0 \\
EU &= \begin{cases} E_m & WU \geq E_m \\ WU & WU < E_m \end{cases} \\
EL &= \frac{WL}{WL_m}(E_m - EU) \\
E &= EU + EL
\end{aligned}
\tag{1}
$$

where, $E_m$ is evaporation, $k_c$ is evaporation conversion coefficient, $E_0$ is evaporation of evaporating dish, $EU$ is soil evaporation in the upper layer, and $EL$ is soil evaporation in the lower layer. Evapotranspiration loss was deducted in the calculation of production and convergence.

### 2.3. Runoff Module

2.3.1. Dominant Runoff Processes

Dominant runoff process (DRP) refers to the runoff model that contributes the most to a rainfall–runoff process. In the present study, the spatial distribution of the dominant runoff process was comprehensively analyzed in combination with underlying surface conditions, factors such as topography, geology, and land utilization that can affect the environment. Soil infiltration is a factor directly affecting regional runoff mechanisms [18] and, for the soil category with poor permeability, we are able to determine its permeability performance according to its properties. It is evident that infiltration excess is the primary runoff process; however, the particular runoff processes of farmland, grassland, forest land, and other types should be determined according to their respective slope and vegetation coverage. Generally, the dominant runoff model of town defaults to infiltration excess runoff; however, near the river, where soil can maintain moisture for longer, saturation excess runoff often occurs [21].

Check dams are set up along the river, their reservoirs boasting long-term water storage, high soil moisture content and prolonged moistness. As the amount of muck rises, the gradient of the reservoir area will gradually level off. Consequently, saturation excess runoff is likely to be the main runoff process in the reservoir area. During flood season, terraces can hold back floodwaters, reducing peak flow, increasing soil water content and increasing river runoff in non-flood seasons. Experiments have also confirmed that saturation excess runoff is the primary runoff process of terraces. The slope shows the steepness of the ground, which directly affects the runoff retention time and infiltration volume. In areas with a gentle gradient, runoff can obtain a longer confluence time to increase soil infiltration and reduce surface runoff; in steep gradient areas, due to the rapid movement of surface runoff and low infiltration, the runoff will be high. When the slope of forest land is less than 3° and the slope of grassland and cultivated land is less than 5°, the soil can quickly reach saturation and become full during the runoff.

Increasing vegetation coverage leads to more rainfall interception by plants, and the increase in soil infiltration capacity will improve the water storage capacity of the soil, making it more prone to become saturated runoff. The vegetation coverage is calculated by the binary model. It considers the unit vegetation coverage to be composed of the weighted area proportion of the vegetation coverage part and the non-vegetation coverage part. Taking 40% and 70% as the classification points, this is divided into three categories: low vegetation coverage (FVC $\leq$ 40%), medium vegetation coverage (40% $\leq$ FVC $\leq$ 70%), and high vegetation coverage (FVC $\geq$ 70%). Only when the vegetation coverage is greater than 40% might the saturation excess runoff model occur. According to the above rules, each underlying surface factor layer is superimposed on the ArcGIS 10.4 platform, and the spatial distribution of the watershed runoff pattern can be obtained by performing the discrimination rules (Figure 2). The identification of the dominant flow generation mechanism is used to select the flow generation model.

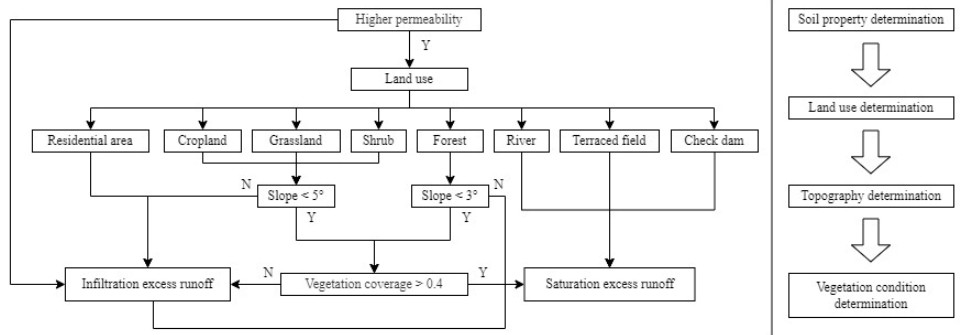

**Figure 2.** Discriminant rules of dominant runoff pattern in watershed.

### 2.3.2. Runoff Calculation

The runoff generation model of the basin will be influenced by various factors on the underlying surface, such as terrain, land use, vegetation cover, etc., resulting in different combinations of runoff generation mechanisms. According to the aeration zone structure and meteorological conditions in different regions, there may be nine types of runoff generation mechanisms, which can be divided into two types according to the types of factors affected, namely the overpermeability and full accumulation runoff generation modes. Due to the spatio-temporal complexity of the underlying surface conditions and meteorological conditions in the catchment, it is impossible to obtain a completely uniform distribution. Therefore, the runoff generation modes at each point in the catchment are different, and the overpermeable and full flow modes are often intertwined. Therefore, the actual runoff generation mode in the basin is the form of the combination of different runoff generation mechanisms in each unit and will change with changes in the underlying surface environment and with the spatio-temporal change of meteorological conditions.

The model uses the infiltration capacity distribution curve to calculate the excess infiltration flow. The average infiltration rate curve of all points in the catchment, under conditions of adequate water supply, is called the catchment infiltration capacity curve. Horton's formula for infiltration capacity has been adopted:

$$f = f_c + (f_0 - f_c)e^{-kt} \tag{2}$$

where, $f$ is the infiltration capacity at time $t$ (mm/h), $f_c$ is the stable infiltration rate (mm/h), $f_0$ is the initial infiltration capacity (mm/h) and $k$ is the index related to the soil permeability characteristics ($h^{-1}$).

The model uses the watershed storage capacity distribution curve to calculate the saturation excess flow. The water storage capacity at different points in the basin varies with the underlying surface conditions. The water storage capacity distribution curve takes into account the influence of the water storage capacity on the flow production at different points in the basin. Based on experience, the shape of the $n$-th parabola is as follows:

$$\alpha = 1 - (1 - \frac{W'}{W'_m})^n \tag{3}$$

where, $W'$ is water storage capacity (mm); $W'_m$ is maximum point water storage capacity (mm) of the basin; $\alpha$ is relative area, which represents the ratio of area to basin area; and $n$ is empirical index.

### 2.4. Confluence Module

Watershed confluence is a complex process, and the complexity of water storage and discharge is exacerbated by the law of soil and water conservation. Mathematical formulas are inadequate to determine the amount of water that can be stored and when it should be discharged for terrace or for medium-sized check dams. This renders physical mechanisms-based confluence computations insufficiently realistic. At present, machine learning research on flood simulation has just begun. Most of the research is directly based on rainfall input to calculate the outlet flow of the watershed, but no attempt has been made to calculate the confluence. Here we used LSTM for confluence calculation.

LSTM is a special kind of RNN, which can maintain the information of the previous input and participate in the resulting output all the time, without gradient disappearance. Therefore, it is more suitable for calculating data with long-term correlation than traditional RNNs. Compared with RNN, wherein memory cell units are added to the hidden layer, LSTM substitutes the RNN cell units, thus enabling it to extract features from temporal sequence data (Figure 3).

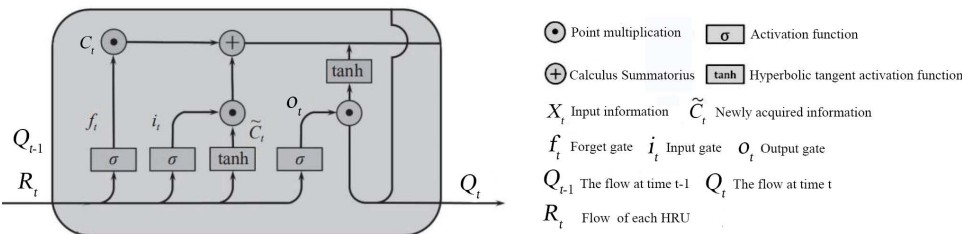

**Figure 3.** Schematic diagram of LSTM calculus.

The flow rate $R_t$ of each HRU at t time and the flow rate $Q_{t-1}$ of outlet section flow at time $t-1$ will be used as the input at time $t$, and the outlet flow $Q_t$ at time $t$ will be output after a series of operations. The specific calculation process is as follows:

$$
\begin{aligned}
f_t &= \sigma\Big[W_f(Q_{t-1}, R_t) + b_f\Big] \\
i_t &= \sigma\Big[W_f(Q_{t-1}, R_t) + b_i\Big] \\
\widetilde{C}_t &= \tanh[W_c(Q_{t-1}, R_t) + b_c] \\
C_t &= f_t \odot C_{t-1} + i_t \odot \widetilde{C}_t \\
o_t &= \sigma[W_0(Q_{t-1}, R_t) + b_0] \\
Q_t &= o_t \odot \tanh(C_t)
\end{aligned}
\tag{4}
$$

where, $W_f$ is the weight matrix of the input gate, $b_f$ is the offset vector of the input gate, $C_t$ is the state variable, and $b_c$ is the offset vector of $\widetilde{C}_t$. We ran the LSTM model with python, dividing 25,652 HRU into catchments and utilizing 29 flood data. This provided us with an adequate amount of data necessary for training.

### 2.5. Soil Erosion Module

Soil erosion is most directly related to runoff through runoff movement and transport in event-based rainfall [32]. Because the CSLE model exploited by Liu et al. (2002) [33] can only be used to calculate annual soil loss, it cannot be used to estimate soil loss in a watershed during an event-based rainfall. Thus, Shi et al. (2018) [34] proposed a CSLE model based on event-based rainfall in which other underlying surface condition factors are retained and annual rainfall and runoff erosion factor are replaced with the $\alpha EkI30$ of the considered runoff depth, rainfall intensity, and rainfall kinetic energy. The specific calculation formula is as follows:

$$
A = x(\alpha E_k I_{30})^y \times K \times L \times S \times B \times E \times T
\tag{5}
$$

$$
E_k = \sum_{t=1}^{n} (e_t v_t)
\tag{6}
$$

$$
e_t = 0.29[1 - 0.72 \exp(-0.05 t_r)]
\tag{7}
$$

where, $A$ is the soil loss of individual rainfall (t·km$^{-2}$); $\alpha$ is the runoff coefficient, which is calculated from the runoff module; $E_k$ (MJ·ha$^{-1}$) is the total kinetic energy of the rainfall, and the maximum 30 min rainfall intensity $I_{30}$ (mm·h$^{-1}$); $e_t$, $v_t$ and $t_r$ are the unit rainfall energy (MJ·ha$^{-1}$·mm$^{-1}$), rainfall (mm), and rainfall intensity (mm·h$^{-1}$) within the rainfall time interval $r$, respectively; $x$ and $y$ are empirical coefficients; $K$ is erodibility factor (t hm$^2$ h hm$^{-2}$ MJ$^{-1}$ mm$^{-1}$); $L$ is slope length factor; $S$ is slope factor; $B$ is vegetation coverage and biological measures; $E$ is engineering measure factor; and $T$ is tillage practice factor. The soil erosion module was used to calculate sediment yield.

### 2.6. Model Parameters and Model Evaluation

There are 10 parameters to be calibrated in the model, including three evapotranspiration parameters, five runoff parameters, and two sediment parameters (Table 1).

**Table 1.** Main parameters of the model.

| Modules | Parameters | Meaning | Range |
|---|---|---|---|
| Evapotranspiration parameters | $WUM$ | Upper water storage capacity of watershed | 24.13–49.97 mm |
| | $WLM$ | Water storage capacity of lower watershed | / |
| | $K_c$ | Conversion coefficient of evaporation | / |
| Runoff yield parameters | $WM$ | Watershed storage capacity | 88–157 mm |
| | $m$ | Empirical index of infiltration capacity curve | / |
| | $n$ | Empirical index of storage capacity curve | / |
| | $f_c$ | Stable infiltration rate | 2.597–8.760 mm |
| | $k$ | Permeability coefficient | 0.158–0.295 |
| Sediment yield parameters | $x$ | Sediment yield empirical coefficient | / |
| | $y$ | Sediment yield empirical coefficient | / |

The relative error of peak flow ($RE_Q$), relative error of runoff depth ($RE_R$), relative error of soil erosion ($RE_A$) and Nash efficiency coefficient ($NSE_Q$) were selected as the evaluation criteria for the accuracy evaluation of the simulation results.

## 3. Case Study

### 3.1. Overview of the Study Area

The Chenggou River basin is situated in the Anding District of Dingxi City, Gansu Province in Northwest China, and is a part of the fifth sub-region of the loess hilly and gully region. It is a tributary of the Zuli River system in the Yellow River basin, with a total area of 161.37 km². Its geographical coordinates range from 104°14′15″–104°28′31″ E, 35°41′7″–35°35′10″ N, and the altitude varies from 1957–2273 m. It has a temperate semi-arid climate, with an average temperature of 6.3 °C and an average annual precipitation of 380 mm (1980–2020). The majority of the rain falls during July to September (67%) and is mostly of the rainstorm type. Annual evaporation reaches 1500 mm. In 2006, the Chenggou River basin was included in the initial group of monitoring projects of the small watershed dam system in Loess Plateau, resulting in the construction of 74 check dams, including 20 backbone dams, 22 medium dams, and 32 small dams. By the end of 2008, all check dams had been completed, with no new check dams having been constructed since but with the area under control of check dams still making up 54.80% of the watershed. In 2015, land use included terraced fields (47.35%), forests (0.54%), grasslands (10.78%), shrubs (16.24%), cultivated land (20.76%), water surface (0.1%) and residential areas (0.73%) of the total area (Figure 4).

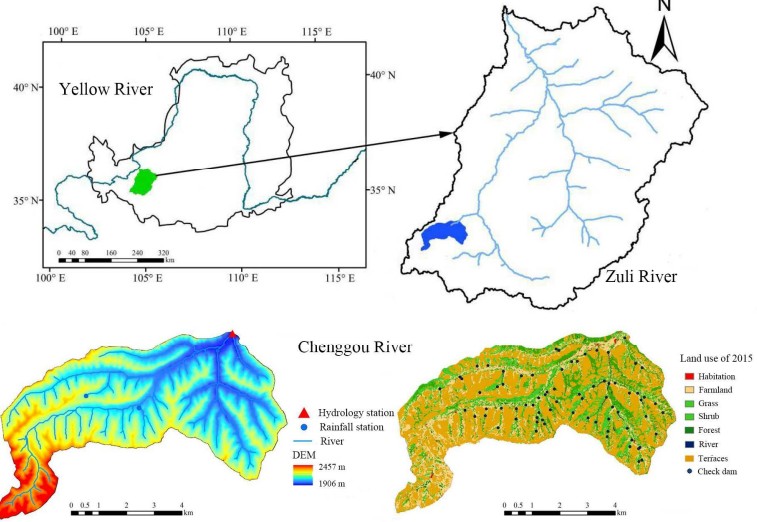

**Figure 4.** Study area.

### 3.2. Dataset

(1)    Hydrometeorological data

The data of the Chenggou River basin include the runoff data from the hydrology station for the period 2013 to 2017, as well as the rainfall data of the three rainfall stations of Gaojiacha, Bieduchuan, and Yangshanzui in the watershed. To make the rainfall–runoff data correspond across time, it was interpolated to a 1 h time step.

(2)    Geospatial data

The remote sensing image uses the Landsat TM (1983–2020) of the United States, with a spatial resolution of 30 × 30 m. The watershed DEM has a resolution of 30 m and is generated from geospatial data. The soil type map provides the geographic distribution of soil types in the watershed, based on Chinese soil genetic classification and linked to FAO's revised legend.

### 3.3. Model Construction

#### 3.3.1. Selection of Flood Events

The study selected 29 relatively complete 1-h meteorological and hydrological data as the basis for model simulation applications (Table 2), of which the first 22 flood events were used as calibration periods and the last seven flood events as validation periods.

**Table 2.** Statistics of selected flood events in Chenggou River basin from 2013 to 2017.

| Item | Flood Times | Duration of Flood (h) | Rain Fall (mm) | Run-Off Depth (mm) | Peak Flow ($m^3$/s) | Soil Loss Amount (t/km$^2$) |
|---|---|---|---|---|---|---|
| | 2013.07.07 | 62 | 15.73 | 0.34 | 0.26 | 0.109 |
| | 2013.07.14 | 29 | 11.195 | 0.27 | 0.29 | 0.087 |
| | 2013.08.06 | 23 | 24.91 | 0.37 | 0.61 | 0.413 |
| | 2013.08.20 | 28 | 17.3 | 0.34 | 0.62 | 0.304 |
| | 2014.04.16 | 25 | 14.7 | 0.24 | 0.31 | 0.220 |
| | 2014.06.18 | 25 | 57.759 | 2.23 | 2.6 | 19.320 |
| | 2014.06.28 | 30 | 14.846 | 0.31 | 0.27 | 0.109 |
| | 2014.07.08 | 30 | 33.26 | 0.54 | 0.64 | 0.739 |
| | 2014.08.21 | 33 | 23.645 | 0.36 | 0.5 | 0.330 |
| | 2014.09.21 | 40 | 20.246 | 0.21 | 0.25 | 0.065 |
| | 2014.10.10 | 45 | 20.75 | 0.34 | 0.45 | 0.087 |
| Calibration period | 2015.05.20 | 30 | 16.3 | 0.32 | 0.6 | 0.500 |
| | 2015.05.27 | 44 | 30.38 | 0.6 | 0.56 | 0.348 |
| | 2015.06.22 | 28 | 25.36 | 0.3 | 0.46 | 0.270 |
| | 2015.07.02 | 33 | 14.5 | 0.46 | 0.48 | 0.543 |
| | 2015.07.08 | 40 | 42.81 | 0.99 | 1.03 | 1.250 |
| | 2015.08.03 | 37 | 34.35 | 2.02 | 1.97 | / |
| | 2016.06.15 | 29 | 17.3 | 0.33 | 0.54 | 0.348 |
| | 2016.07.10 | 36 | 15.4 | 0.2 | 0.58 | 0.196 |
| | 2016.07.18 | 30 | 21.4 | 0.5 | 0.68 | 0.826 |
| | 2016.08.24 | 35 | 23.8 | 0.27 | 0.83 | 0.304 |
| | 2016.09.10 | 28 | 17.63 | 0.3 | 0.52 | 0.239 |
| | 2017.05.02 | 31 | 20.3 | 0.39 | 0.58 | 0.304 |
| | 2017.05.21 | 32 | 19.7 | 0.22 | 0.33 | 0.109 |
| | 2017.06.19 | 29 | 18.9 | 0.35 | 0.45 | 0.304 |
| Verification period | 2017.07.04 | 32 | 24.6 | 0.89 | 1.3 | 0.783 |
| | 2017.08.03 | 38 | 20.8 | 0.8 | 1.2 | 1.848 |
| | 2017.08.06 | 41 | 36.64 | 2.69 | 2.08 | 4.600 |
| | 2017.08.27 | 59 | 38.1 | 0.75 | 1.06 | 0.457 |

#### 3.3.2. Spatial Analysis of Runoff Generation Mode

A total of 47,796 HRUs were divided in the Chenggou River basin (Figure 5). The saturated excess runoff HRUs totaled 13,731, making up 28.73%. The total infiltration

excess runoff was 34,065, making up 71.27% of the total. Of the various land use types, terraces had 1548 HRUs and check dams had 258. Check dams and terraces accounted for 3.78% of the total HRUs in the watershed. The area of terraces accounts for a large proportion, and the HRUs area of the runoff generation model is 52.95% of the whole watershed. In addition to the terraces, the HRUs of the other runoff generation models are distributed along the river. The check dams in the watershed are mainly distributed in the east, with the runoff in the southeast of the watershed especially apt to be intercepted by the check dams.

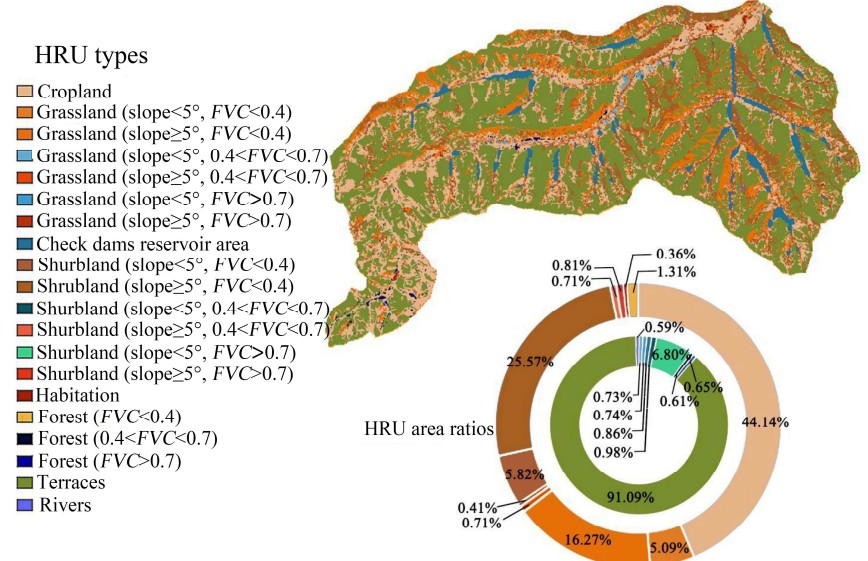

**Figure 5.** Division of HRU in Chenggou River basin.

### 3.3.3. Model Parameter Calibration

HRU types with different characteristics of underlying surface factors need to calibrate different parameters due to the differences in their runoff–sediment mechanisms. Generally speaking, check dams do not discharge water when if it does not reach the designed discharge capacity, so flood calculation is based on the premise that the discharge conditions are not met; that is, that the check dam can intercept all the water and sediment in its control area. In order to ensure the accuracy of the confluence process, the number of hidden layer neurons is set to 50 in the training of the LSTM module, the number of training times per neuron is set to 150, while the input data is the runoff depth generated hourly by HRU. There are 10 parameters in the overall model structure, divided into three categories (Tables 3 and 4 and Figure 6).

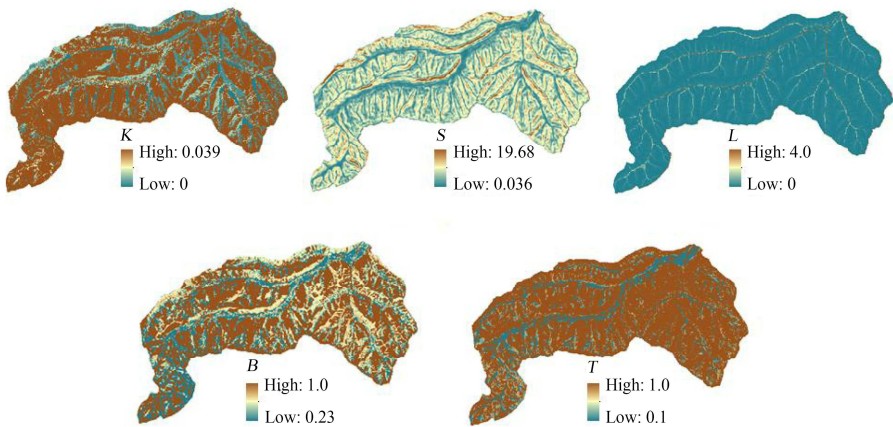

**Figure 6.** Spatial distribution of sediment yield parameters in Chenggou River.

**Table 3.** Calibration results of HRU parameters in the Chenggou River basin.

| HRU Type | Dominant Runoff Process Type | WM | WUM | WLM | $K_c$ | $f_c$ |
|---|---|---|---|---|---|---|
| Cultivated land | Infiltration excess runoff | 153 | 28 | 125 | 0.46 | 5.3 |
| Grass land (gradient < 5°, *FVC* < 40%) | Infiltration excess runoff | 160 | 30 | 130 | 0.33 | 4.4 |
| Grass land (gradient ≥ 5°, vegetation coverage < 40%) | Infiltration excess runoff | 160 | 30 | 130 | 0.33 | 5.3 |
| Grass land (gradient < 5°, vegetation coverage ≥ 40%) | Saturation excess runoff | 152 | 29 | 123 | 0.42 | 5.3 |
| Grass land (gradient ≥ 5°, vegetation coverage ≥ 40%) | Infiltration excess runoff | 152 | 29 | 123 | 0.42 | 5.9 |
| Woodland (vegetation coverage ≤ 40%) | Infiltration excess runoff | 138 | 20 | 118 | 0.42 | 5.3 |
| Woodland (40% < vegetation coverage ≤ 70%) | Saturation excess runoff | 133 | 20 | 113 | 0.46 | 6.1 |
| Woodland (vegetation coverage ≥ 70%) | Saturation excess runoff | 126 | 20 | 106 | 0.51 | 7.1 |
| Terrace | Saturation excess runoff | 143 | 27 | 116 | 0.47 | 5.5 |
| Grass land (gradient < 5°, vegetation coverage < 40%) | Infiltration excess runoff | 145 | 25 | 127 | 0.40 | 5.9 |
| Grass land (gradient ≥ 5°, vegetation coverage < 40%) | Infiltration excess runoff | 145 | 25 | 127 | 0.4 | 5.3 |
| Grass land (gradient < 5°, vegetation coverage ≥ 40%) | Saturation excess runoff | 133 | 23 | 118 | 0.46 | 5.3 |
| Grass land (gradient ≥ 5°, vegetation coverage ≥ 40%) | Infiltration excess runoff | 133 | 23 | 118 | 0.46 | 6.1 |
| Residential district | Infiltration excess runoff | 155 | 25 | 130 | 0.40 | 5.3 |

**Table 4.** Calibration results of other parameters in the Chenggou River basin.

| Parameter | $k$ | $m$ | $n$ | $x$ | $y$ |
|---|---|---|---|---|---|
| 2013~2017 | 0.251 | 0.39 | 0.11 | 1.03 | 0.812 |

3.3.4. Model Verification

The average relative error of runoff depth of 29 floods is 8.86%. Among these, the relative error of runoff depth of 24 of the floods is less than 10%, and the relative error of runoff depth of five of the floods is between 10% and 20%. The relative error of average peak flow is 9.44%. Among these, the relative error of average peak flow of 14 of the floods is less than 10%, of 13 of the floods it is more than 10%, and of one flood is more than 20%. The Nash efficiency coefficient $NSE_Q$ of 29 flood events is 0.84. The relative error of soil erosion is 14.45%, and there are no data for the sediment side leakage accident of August 3rd, 2015. Among these, the relative error of soil erosion in 11 of the floods is less than 10%, the relative error of soil erosion in 12 of the floods is more than 10%, and the relative error of soil erosion in five of the floods is more than 20% (Table 5). Some flood event hydrographs are shown in Figure 7, and the rest are shown in Supplementary Materials, Figure S1.

**Table 5.** Simulation results of 29 floods in the Chenggou River basin from 2013 to 2017.

| Flood Numbers | Actual RunOff Depth (mm) | Simulated Run-Off Depth (mm) | $RE_R$ (%) | Actual Peak Flow (m³/s) | Simulated Flood Peak Flow (m³/s) | $RE_Q$ (%) | Actual Soil Erosion (t/ha) | Simulated Soil Erosion (t/ha) | $RE_A$ (%) |
|---|---|---|---|---|---|---|---|---|---|
| 2013.07.07 | 0.34 | 0.36 | 5.88% | 0.26 | 0.28 | 7.69% | 0.109 | 0.136 | 25.32% |
| 2013.07.14 | 0.27 | 0.28 | 3.70% | 0.29 | 0.33 | 13.79% | 0.087 | 0.094 | 8.32% |
| 2013.08.06 | 0.37 | 0.34 | −8.11% | 0.61 | 0.68 | 11.48% | 0.413 | 0.375 | −9.27% |
| 2013.08.20 | 0.34 | 0.36 | 5.88% | 0.62 | 0.69 | 11.29% | 0.304 | 0.334 | 9.77% |
| 2014.04.16 | 0.24 | 0.22 | −8.33% | 0.31 | 0.29 | −6.45% | 0.220 | 0.274 | 24.41% |
| 2014.06.18 | 2.23 | 2.29 | 2.69% | 2.6 | 2.4 | −7.69% | 19.320 | 17.014 | −11.93% |
| 2014.06.28 | 0.31 | 0.33 | 6.45% | 0.27 | 0.29 | 7.41% | 0.109 | 0.095 | −12.29% |

**Table 5.** *Cont.*

| Flood Numbers | Actual RunOff Depth (mm) | Simulated Run-Off Depth (mm) | $RE_R$ (%) | Actual Peak Flow (m³/s) | Simulated Flood Peak Flow (m³/s) | $RE_Q$ (%) | Actual Soil Erosion (t/ha) | Simulated Soil Erosion (t/ha) | $RE_A$ (%) |
|---|---|---|---|---|---|---|---|---|---|
| 2014.07.08 | 0.54 | 0.58 | 7.41% | 0.64 | 0.6 | −6.25% | 0.739 | 0.632 | −14.43% |
| 2014.08.21 | 0.36 | 0.34 | −5.56% | 0.5 | 0.53 | 6.00% | 0.330 | 0.268 | −18.76% |
| 2014.09.21 | 0.21 | 0.19 | −9.52% | 0.25 | 0.28 | 12.00% | 0.065 | 0.083 | 28.03% |
| 2014.10.10 | 0.34 | 0.31 | −8.82% | 0.45 | 0.53 | 17.78% | 0.087 | 0.094 | 8.32% |
| 2015.05.20 | 0.32 | 0.35 | 9.37% | 0.6 | 0.54 | −10.00% | 0.500 | 0.431 | −13.89% |
| 2015.05.27 | 0.6 | 0.66 | 10.00% | 0.56 | 0.53 | −5.36% | 0.348 | 0.382 | 9.83% |
| 2015.06.22 | 0.3 | 0.32 | 6.67% | 0.46 | 0.49 | 6.52% | 0.270 | 0.225 | −16.77% |
| 2015.07.02 | 0.46 | 0.48 | 4.35% | 0.48 | 0.54 | 12.50% | 0.543 | 0.579 | 6.50% |
| 2015.07.08 | 0.99 | 1.1 | 11.11% | 1.03 | 1.1 | 6.80% | 1.250 | 1.134 | −9.29% |
| 2015.08.03 | 2.02 | 2.21 | 9.41% | 1.97 | 2.19 | 11.17% | / | 7.562 | / |
| 2016.06.15 | 0.33 | 0.35 | 6.06% | 0.54 | 0.51 | −5.56% | 0.348 | 0.410 | 17.86% |
| 2016.07.10 | 0.2 | 0.17 | −15.00% | 0.58 | 0.52 | −10.34% | 0.196 | 0.170 | −13.25% |
| 2016.07.18 | 0.5 | 0.48 | −4.00% | 0.68 | 0.62 | −8.82% | 0.826 | 0.658 | −20.36% |
| 2016.08.24 | 0.27 | 0.25 | −7.41% | 0.83 | 0.72 | −13.25% | 0.304 | 0.246 | −19.30% |
| 2016.09.10 | 0.3 | 0.26 | −13.33% | 0.52 | 0.47 | −9.62% | 0.239 | 0.283 | 18.46% |
| 2017.05.02 | 0.39 | 0.36 | −7.69% | 0.58 | 0.63 | 8.62% | 0.304 | 0.293 | −3.70% |
| 2017.05.21 | 0.22 | 0.2 | −9.09% | 0.33 | 0.38 | 15.15% | 0.109 | 0.125 | 14.73% |
| 2017.06.19 | 0.35 | 0.32 | −8.57% | 0.45 | 0.4 | −11.11% | 0.304 | 0.310 | 1.74% |
| 2017.07.04 | 0.89 | 0.97 | 8.99% | 1.3 | 1.42 | 9.23% | 0.783 | 0.662 | −15.37% |
| 2017.08.03 | 0.8 | 0.89 | 11.25% | 1.2 | 1.05 | −12.50% | 1.848 | 1.711 | −7.38% |
| 2017.08.06 | 2.69 | 2.75 | 2.23% | 2.08 | 2.3 | 10.58% | 4.600 | 5.772 | 25.48% |
| 2017.08.27 | 0.75 | 0.72 | −4.00% | 1.06 | 1.28 | 20.75% | 0.457 | 0.434 | −5.00% |

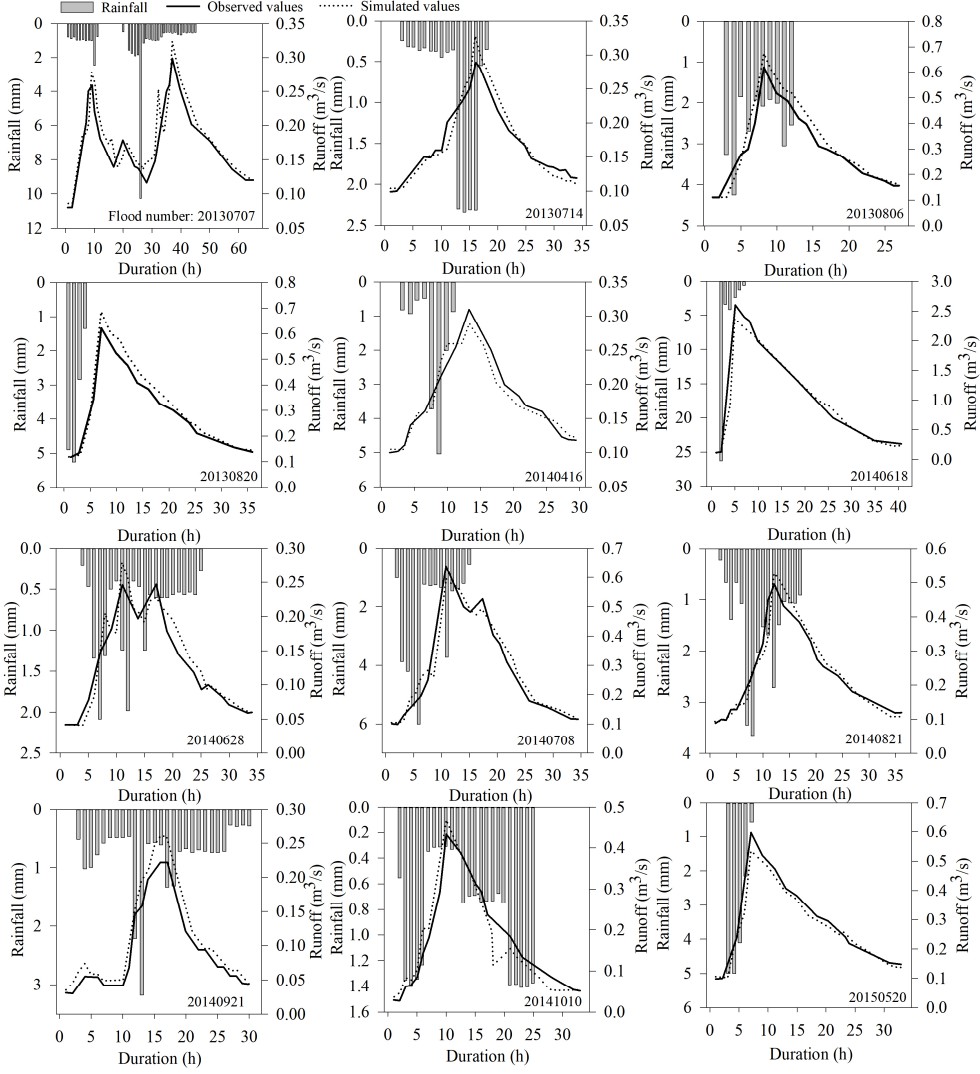

**Figure 7.** Comparison of simulated and observed flood runoff in some events.

## 4. Discussions

The runoff pattern of the Loess Plateau is shifting, with infiltration excess runoff proportion decreasing and saturation excess and mixed runoff proportion increasing [35]. This change is closely related to the ecological construction project of the Loess Plateau. Hu et al. (2020b) [36] selected 60 rain–flood data from 1966 to 2006 in the Jialu River basin of the Loess Plateau, and comprehensively analyzed the response mechanism of the watershed runoff mechanism to the change of forest and grass cover. The results show that the proportion of infiltration excess runoff mechanisms decreased, the proportion of interflow and saturation excess runoff mechanisms increased gradually, and the water storage capacity of the watershed increased with the increase of vegetation cover. Similarly, under the influence of human activities, the decrease of peak flow, the delay of peak time, and the extension of recession time also occurred in the upper reaches of Fenhe River, Zuli River, Daqing River, and Xichuan River [37–39]. Due to the change of influencing factors in the process of runoff production, the dominant runoff mechanism of the watershed may change. When simulating runoff in semi-arid areas, considering both infiltration excess runoff and saturation excess runoff may produce better results. For example, Liu et al. (2012) [40] combined their simulation with the Tsinghua hydrological model and found that the dominant runoff mechanism in the Weihe River basin may transition from over-infiltrating surface runoff to saturated surface runoff with the move from point scale to surface scale. Hu et al. (2014) [41] used the Simplex, Rosenbrock, and Genetic methods to improve the compatibility model of saturation and infiltration, turning it into a three-source model and verifying it in the Yihe River basin. By comparing the simulation results of the compatibility model with that of the improved combined model, it is evident that the latter can better reflect the change process of various runoff components and more accurately reflect reality. Consequently, it is useful to investigate the impact of check dams and terraces on runoff and sediment in the Loess Plateau and necessary to start with the underlying surface mechanism. Therefore, this study started from the underlying surface of the watershed, considers factors such as land use, vegetation coverage, and topography, and established a discriminant rule for the dominant runoff pattern of the watershed. The key to the refined expression of the runoff process is to realize the classification simulation of the runoff process.

With the gradual improvement of supporting technologies such as DEM, GIS, and RS, distributed hydrological models have developed rapidly, especially the widespread use of DEM. At present, most of the distributed hydrological models rely on the grid structure of DEM to extract the underlying surface information and design the calculation structure of the model [42,43]. However, in the Loess Plateau region, a series of engineering measures that were built to prevent soil erosion, such as check dams, terraces, and strip fields, have greatly changed the original topography and the water path. The digitally generated DEM, based on the topographic maps drawn in the 1960s, or the shared DEM data downloaded from ASTER, Japan, cannot all represent the current real terrain and this brings difficulties to the construction and operation of the distributed hydrological model, especially the calculation of the confluence process. Due to the lack of description of underlying surface characteristics, aquifer structure, and some hydrological processes, the physical hydrological model has some errors in the simulation of the runoff process. The deep learning model is directly based on historical observation data to train and fit the runoff process. Due to its strong data mining and fitting ability, the deep learning model will help to promote the understanding of hydrological processes and improve the simulation accuracy of physical hydrological models. In addition, the spatial heterogeneity of the underlying surface in the study area is obvious, and the mechanism of runoff generation and confluence is quite different. Therefore, the spatial distribution of precipitation has a significant impact on the runoff high-water process. As can be seen from Figure 7, in the case of high water and sediment, the model simulation may be high. The author believes that the reasons may found in be the following two points: (1) the model parameters are still in error and (2) the interception effect of the terrace is not obvious. The terrace alters the

runoff mechanism of the surfaces beneath it and maximizes the water that both it, and the upstream areas, retain. Under high flow conditions, the difference caused by interception may be stronger, resulting in an overestimation of runoff by the model.

## 5. Conclusions

In recent years, the underlying surface of the Loess Plateau has changed dramatically, affecting the path of the hydrological cycle and causing difficulties in the calculation of runoff production and confluence. The confluence calculation based on machine learning can effectively avoid the physical change of the confluence process caused by the change of the underlying surface conditions. Based on the characteristics of the underlying surface, this study constructed a runoff–sediment model including traditional physical mechanisms and deep learning. Based on the runoff process dominating the watershed's surface, and combined with a number of other studies, we proposed a basis for the division of the underlying surface runoff model, including check dams and terraces, and the hydrological response unit (HRU) of the watershed, and its area was thus divided into two dominant runoff modes: saturation excess runoff and infiltration excess runoff. For these two runoff models, the infiltration capacity distribution curve and the watershed storage capacity distribution curve were selected to calculate the runoff yield of each HRU. Considering the impact of soil and water conservation measures on the confluence process, the deep learning module LSTM was used for confluence training in order to find the confluence characteristics of the watershed, and the confluence calculation was carried out according to the runoff yield of each HRU. The CSLE model based on individual rainfall was used to calculate the amount of soil erosion caused by each rainfall, considering the slope, slope length, soil, biological measures, tillage measures, and engineering measures. The model performed well in 29 flood events simulation.

The model considers the division of watershed hydrological response unit mainly through the underlying surface factors and calibrates different parameters for each type of hydrological response unit. As the number of considered underlying surface factors increases, the type of hydrological response unit will increase explosively, making it more difficult to simulate large watersheds. This study may provide a new reference for the calculation of production and confluence in the Loess Plateau.

**Supplementary Materials:** The following supporting information can be downloaded at: https://www.mdpi.com/article/10.3390/su15064894/s1, Figure S1: Comparison of simulated and observed flood runoff values.

**Author Contributions:** Conceptualization, Y.T. and S.J.; methodology, Y.T.; software, P.X.; validation, P.J. and P.X.; formal analysis, Y.T. and S.J.; investigation, P.X.; resources, P.J.; data curation, S.J.; writing—original draft preparation, P.J. and S.J.; writing—review and editing, Y.T.; visualization, P.X.; supervision, P.X.; project administration, S.J.; funding acquisition, S.J. All authors have read and agreed to the published version of the manuscript.

**Funding:** This research was funded by Yellow River Science Joint Foundation of National Natural Science Foundation (U2243210), the Open Project Foundation of Key Laboratory of Ministry of Water Resources Soil and Water Loss on Loess Plateau (WSCLP202203), Training Program for Young Backbone Teachers in Colleges and Universities of Henan Province (2021GGJS003), the Henan Natural Science Foundation (212300410413), the Henan Youth Talent Promotion Project (2021HYTP030).

**Institutional Review Board Statement:** Not applicable.

**Informed Consent Statement:** Not applicable.

**Data Availability Statement:** Not applicable.

**Conflicts of Interest:** The authors declare no conflict of interest.

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
