# Peer review of "Runoff–Sediment Simulation of Typical Small Watershed in Loess Plateau of China"

_sustainability, doi:10.3390/su15064894_

Round 1

Reviewer 1 Report

Dear Authors, that is good job. Please update your MS according to a few explanations add on MS. Sincerely.

Author Response

Reviewer #1

  1. Add name of location of Yellow River (for exampel, Yellow River, ...(province), China or The river that arises from X and empties into the X sea.

A: we had revised the sentence as follow: The problem of runoff-sediment in the Yellow River, Gansu province, China has always been a concern.

  1. Supporting this information by reference will strengthen your claims.

A: we had added the references in this section.

  1. Check the formulas one more time.

A: Thank you very much, we had revised the formulas.

  1. Please identify, K, L, S, B, E and T. Some of those like RUSLE components

A: K is erodibility factor (t hm2 h hm-2 MJ-1 mm-1); L is slope length factor, S is slope factor, B is vegetation coverage and biological measures; E is engineering measure factor; T is tillage practice factor.

  1. In this section, talk about the advantages of the deep learning model you use over the others. Make some suggestions for the Yellow River and The Plateau.

A: we had revised the conclusions as follow:

In recent years, the underlying surface of the Loess Plateau has changed dramatically, affecting the path of the hydrological cycle and causing difficulties in the calculation of runoff production and confluence. The confluence calculation based on machine learning can effectively avoid the physical change of the confluence process caused by the change of the underlying surface conditions. Based on the characteristics of the underlying surface, this study constructed a runoff-sediment model including traditional physical mechanisms and deep learning. Based on the dominant runoff process of the underlying surface of the watershed, combined with a number of studies, proposed the division basis of the underlying surface runoff model including check dams and terraces, divided and the hydrological response unit (HRU) of the watershed,and its area is divided into two dominant runoff modes: saturation excess runoff and infiltration excess runoff. For these two runoff models, the infiltration capacity distribution curve and the watershed storage capacity distribution curve are selected to calculate the runoff yield of each HRU. Considering the impact of soil and water conservation measures on the confluence process, the deep learning module LSTM is used for confluence training to find the confluence characteristics of the watershed, and the confluence calculation is carried out according to the runoff yield of each HRU. The CSLE model based on individual rainfall was used to calculate the amount of soil erosion caused by each rainfall, considering the slope, slope length, soil, biological measures, tillage measures, and engineering measures. The model performs well in 29 flood events simulation.

The model considers the division of watershed hydrological response unit mainly through the underlying surface factors and calibrates different parameters for each type of hydrological response unit. As the number of underlying surface factors considered increases, the type of hydrological response unit will increase explosively, making it more difficult to simulate large watersheds. This study can provide a new idea for the calculation of production and confluence in the Loess Plateau.

Reviewer 2 Report

Dear Authors

You have presented a very good manuscript, congratulations. I have some suggestions below to help the reader better connect with your manuscript.

1- The innovation of the research as well as the difference between your research and other researchers should be bolded at the end of the introduction section.

2- Provide more explanations about the used models so that readers can better relate to the manuscript.

3- For each model, explain in what part of your research it was used.

4- Explain about the LSTM model, what is it used for? What were the input parameters? Used in multivariate mode? With what tool is it implemented?

5- Is the use of the LSTM model sufficient for the amount of data used? According to the learning base of this algorithm, does it reach the necessary learning with the amount of data used in the research?

6- There should be more clarification about the research method

Author Response

Reviewer #2

  • The innovation of the research as well as the difference between your research and other researchers should be bolded at the end of the introduction section.

A: We had bolded the following sentences “The main contributions in the present study are as follows: (1) According to the theory of watershed runoff, based on factors such as land use, slope, vegetation cover, and so on, the discriminating mechanism of the dominant runoff process is established to identify the spatial distribution of dominant runoff process in the underlying surface of the watershed, and to determine the reasons for the change of water and sediment process in the study area from the mechanism level; (2) Based on the spatial distribution characteristics of runoff and sediment mechanism under the changing environment, divided the hydrological response unit of the watershed, and the runoff-sediment model based on the conditional distribution of the underlying pad surface is constructed by combining the traditional physical mechanism and deep learning.

  • Provide more explanations about the used models so that readers can better relate to the manuscript.

A: We had revised section 2.3.2 Runoff calculation as follow:

The runoff generation model of the basin will be influenced by various factors on the underlying surface, such as terrain, land use, vegetation cover, etc., resulting in different combinations of runoff generation mechanisms. According to the aeration zone structure and meteorological conditions in different regions, there may be nine types of runoff generation mechanisms, which can be divided into two types according to the types of factors affected, namely over permeability and full accumulation runoff generation modes. Due to the spatio-temporal complexity of the underlying surface conditions and meteorological conditions in the catchment, it is impossible to have a completely uniform distribution. Therefore, the runoff generation modes at each point in the catchment are not the same, and over permeable and full flow are often intertwined. Therefore, the actual runoff generation mode in the basin is the form of the combination of different runoff generation mechanisms in each unit, and will change with the change of the underlying surface environment and the spatio-temporal change of meteorological conditions.

The model uses the infiltration capacity distribution curve to calculate the excess infiltration flow. The average infiltration rate curve of all points in the catchment under conditions of adequate water supply is called the catchment infiltration capacity curve. Horton's formula for infiltration capacity has been adopted:

where, f is the infiltration capacity at time t (mm/h), fc is the stable infiltration rate (mm/h), f0 is the initial infiltration capacity (mm/h) and k is the index related to the soil permeability characteristics (h-1).

The model uses the watershed storage capacity distribution curve to calculate the saturation excess flow.The water storage capacity at different points in the basin varies with the underlying surface conditions. The water storage capacity distribution curve takes into account the influence of the water storage capacity on the flow production at different points in the basin. Based on experience, the shape of the n-th parabola is take:

where, is water storage capacity (mm); is maximum point water storage capacity (mm) of the basin; is relative area, represents the ratio of area to basin area, n is empirical index.

  • For each model, explain in what part of your research it was used.

A: We had added some explanations in the manuscript.  

  • Explain about the LSTM model, what is it used for? What were the input parameters? Used in multivariate mode? With what tool is it implemented?

A: (1) Due to the drastic changes in the underlying surface of the Loess Plateau, it poses great challenges to the calculation of confluence in this region. Here we use LSTM for confluence calculation.

  • The production flowRt of each HRU at time t and the flow Qt-1 of outlet section at time t-1 will be the input at time t, and the outlet flow Qt at time t will be output after a series of calculations.
  • We use the input flow and output flow of each HRU to conduct LSTM training.
  • We run the LSTM model withpython.
  • Is the use of the LSTM model sufficient for the amount of data used? According to the learning base of this algorithm, does it reach the necessary learning with the amount of data used in the research?

A: We divided 25,652 HRU into catchment and utilized 29 flood data. This provided us with an adequate amount of data necessary for training.

  • There should be more clarification about the research method

A: We had already made modifications to the method section in the text.

Reviewer 3 Report

Soil and water conservation measures such as check dams and terraces in the Loess Plateau of China have profoundly changed the water and sediment conditions. How to accurately simulate the runoff-sediment process under complex underlying surface conditions has become the key to clarifying the water cycle law.

In order to prevent soil erosion, a series of water conservation projects have been built on the Loess Plateau. These water conservation projects have effectively reduced soil erosion. But it changes the path of hydrological cycle and brings difficulties to hydrological simulation. In this paper, machine learning is used to simulate the confluence process, which solves the problem well.

At the same time, the author added the judgment combination of flow generation modes in the process of flow generation, and endowed different flow generation modes according to different underlying surface conditions, which is of great significance for improving the simulation of flow generation accuracy. At the same time, a set of method to judge the production pattern is put forward, which is very innovative and can provide a good reference for other related research. There are some points to be revised as follows:

1. Line 29, floods, flood.

2. Line49-51 Focus on the Loess Plateau

3. The relationship between the Yellow River Basin and the Loess Plateau is defined in the introduction.

4. How do slope factors determine runoff patterns.

5. In the future work, it is suggested to use heavy rainfall for verification.

Author Response

Reviewer #3

Soil and water conservation measures such as check dams and terraces in the Loess Plateau of China have profoundly changed the water and sediment conditions. How to accurately simulate the runoff-sediment process under complex underlying surface conditions has become the key to clarifying the water cycle law.

In order to prevent soil erosion, a series of water conservation projects have been built on the Loess Plateau. These water conservation projects have effectively reduced soil erosion. But it changes the path of hydrological cycle and brings difficulties to hydrological simulation. In this paper, machine learning is used to simulate the confluence process, which solves the problem well.

At the same time, the author added the judgment combination of flow generation modes in the process of flow generation, and endowed different flow generation modes according to different underlying surface conditions, which is of great significance for improving the simulation of flow generation accuracy. At the same time, a set of method to judge the production pattern is put forward, which is very innovative and can provide a good reference for other related research. There are some points to be revised as follows:

  1. Line 29, floods, flood.

A: We had revised the word.

  1. Line49-51 Focus on the Loess Plateau

A:We had added the sentence as follow: The Loess Plateau is the major drainage area of the Yellow River basin.

  1. The relationship between the Yellow River Basin and the Loess Plateau is defined in the introduction.

A:We had added the sentence as follow: The Loess Plateau is the major drainage area of the Yellow River basin.

  1. How do slope factors determine runoff patterns.

A: The slope will change the stress on the surface where the rainfall reaches, and will affect the process of precipitation redistribution

  1. In the future work, it is suggested to use heavy rainfall for verification.

A: Thank you very much. It is a good suggestion.

Round 2

Reviewer 2 Report

Dear Authors, 

The answer to the reviewers is well done and the quality of the manuscript is very valuable.

thanks

Author Response

Thank you very much.